# Achieving universal health coverage for people with stroke in South Africa: protocol for a scoping review

Sjan-Mari van Niekerk,[1] Gakeemah Inglis-Jassiem,[1] Sureshkumar Kamalakannan [ID] ,[2,3] Silke Fernandes,[4] Jayne Webster,[5] Rene English,[6] Tracey Smythe [ID] ,[3] QA Louw[1]

► Prepublication history and additional materila for this paper is available online. To view these files, please visit the journal online (http://dx.doi.org/10.1136/bmjopen-2020-041221).

For numbered affiliations see end of article.

**Correspondence to**
Dr Sjan-Mari van Niekerk;
smbrown@sun.ac.za

## ABSTRACT

**Introduction** Stroke is the second most common cause of death after HIV/AIDS and a significant health burden in South Africa. The extent to which universal health coverage (UHC) is achieved for people with stroke in South Africa is unknown. Therefore, a scoping review to explore the opportunities and challenges within the South African health system to facilitate the achievement of UHC for people with stroke is warranted.

**Methods and analysis** The scoping review will follow the approach recommended by Levac, Colquhoun and O'Brien, which includes five steps: (1) identifying the research question, (2) identifying relevant studies, (3) selecting the studies, (4) charting the data, and (5) collating, summarising and reporting the results. Health Systems Dynamics Framework and WHO Framework on integrated people-centred health services will be used to map, synthesise and analyse data thematically.

**Ethics and dissemination** Ethical approval is not required for this scoping review, as it will only include published and publicly available data. The findings of this review will be published in an open-access, peer-reviewed journal and we will develop an accessible summary of the results for website posting and stakeholder meetings.

## INTRODUCTION

Stroke is the second most common cause of death after HIV/AIDS and a significant health burden in South Africa (SA)[1–4] ; an estimated 75 000 strokes occur each year in SA, with a projected burden of disease of 564 000 disability-adjusted life years.[5] The burden of stroke in rural areas of SA is also on the increase with an estimated 33 500 strokes occurring in 2011 in these areas alone.[6] However, the number of people living with stroke in SA is likely underestimated as no national stroke database or registry is in place, and these estimates are calculated from limited studies undertaken in a few parts of the country.

Stroke is the leading cause of disability in adults in SA, placing additional strain on social and healthcare services nationally.[7] Increased (1) prevalence of cardiovascular

### Strengths and limitations of this study

► A comprehensive scoping review methods is proposed to review the question.
► Frameworks such as the Health Systems Dynamics and integrated people-centred health services will be used.
► Global as well as national databases will be searched with comprehensive search strategies.
► The review will be limited to papers published between 2005 and 2020.
► The review will be limited to papers published in English and Afrikaans only.

risk factors (such as hypertension, obesity, diabetes mellitus), (2) unchecked industrialisation and (3) urbanisation, contribute to this epidemiological transition of stroke in many low-income and middle-income countries,[8] including SA.[1] Cardiovascular diseases, including stroke, was previously set to surpass infectious diseases as the major cause of morbidity and mortality in sub-Saharan Africa[9] with the burden of these non-communicable diseases increasing in this region of Africa.[10] In the last decade, a growing body of evidence shows an association with non-communicable diseases and infectious diseases: for example, diabetes mellitus, chronic respiratory conditions and chronic kidney disease have been linked to increased tuberculosis (TB) morbidity and mortality and vice versa.[11] Non-communicable diseases may cause 'impaired immunity, metabolic imbalances and stress factors which favour the manifestation of TB'.[11] Furthermore, younger individuals in their most economically productive years are increasingly being affected in Africa, more often presenting with infectious comorbidities such as HIV/AIDS or TB in addition to stroke,[12 13] which further increases the burden on already strained health systems. With the recent COVID-19

global pandemic, it had become evident that individuals with underlying chronic conditions like stroke are more vulnerable to this infection and might have even more complex health and social care needs.[14]

## Care for people with stroke

The delivery of effective and efficient stroke care is crucial to enhance the physical, cognitive and emotional well-being of individuals post-stroke, to improve their functional independence and quality of life, and ultimately to enable reintegration to their communities.[15 16] For the purposes of this review, we define stroke care as '…the continuum of care starting at the onset of a stroke event through the hyperacute phase, acute inpatient care, stroke rehabilitation, prevention of recurrent stroke, and concludes with community reintegration and long term recovery'.[17] Internationally, delivering effective stroke care has changed over the years as an understanding of stroke has increased, moving from individual care to a more comprehensive health system approach.[18]

The majority of people with stroke in SA may have no or limited access to stroke care and rehabilitation.[3 19] Stroke rehabilitation is limited in the first instance by insufficient rehabilitation facilities in the public sector, as well as inadequate intensity of inpatient rehabilitation, and lack of outpatient or community-based rehabilitation.[20] Reduced access to outpatient community-based stroke rehabilitation services was shown in a predominantly urban province of SA, where people with stroke only received 1–4 hours of physiotherapy sessions with a median number of 1.8 hours over a 6-month period.[21] In SA, stroke care occurs across a range of settings, from tertiary hospitals to remote community primary healthcare facilities; care for people with stroke can be provided individually or in a group setting, at home, in a community environment or a specialist centre.[1] While public health policy in SA ascribes to primary healthcare and a decentralised approach, many stroke care and rehabilitation services remain centralised at district and specialist rehabilitation hospitals,[22] which reinforces the inequality experienced by rural communities in terms of healthcare access.[5] Although home-based care services are available in certain communities, poor referral to and articulation with these services and other healthcare levels, often result in poor service delivery.[23] Individuals who experience mobility limitations post-stroke subsequently experience restricted access to services, as do those with minimal access to transport. Free primary level rehabilitation services are available in some communities, however these services may not be comprehensive or efficient due to shortages of staff skilled in stroke care or incomplete multidisciplinary teams.[22] People with stroke may be discharged home without receiving rehabilitation interventions, and families experience both catastrophic health expenditure and financial loss due to additional responsibilities of caregiving, often leading to an inability to maintain gainful employment.[3 24] As in other low-income and middle-income countries, unmet stroke care and rehabilitation needs are further entrenched due to limited financial and infrastructural/architectural resources.[20 24] In addition, in SA there are no dedicated support systems for caregivers and people with stroke, which is compounded by a population that experience low levels of health literacy.[21 25 26] Health-seeking behaviours may be further affected by the influence of religion and cultural beliefs on patients' views regarding the causes and management of cardiovascular conditions such as stroke, which highlights the need for culturally sensitive stroke care and patient-centric health systems.[27]

## Status of stroke care in SA

The sociopolitical context of SA has had significant effects on national healthcare policies and services, largely due to disparities in social and financial capital based on discriminatory racial and gender divides.[4] Although classified as a middle-income country, SA has high levels of poverty and unemployment, with unemployment rates ranging between 29% (nationally) and 50% (among younger people) and relative poverty worsening over time (Gini coefficient increased from 0.6 in 1995 to almost 0.7 in 2009).[28] These social determinants of health influence the high levels of mortality and morbidity brought about by infectious diseases such as HIV and TB, as well as non-communicable diseases such as diabetes and hypertension.[4 29] Since the first democratic election in 1994, both government and society strive for equality and adopted a human rights-based approach to healthcare reform. Despite well-founded rights-based policy development with practical application, many areas still struggle to achieve equality.[29] The SA government has committed to the WHO vision of achieving equitable, evidence-based rehabilitation for all by 2030,[30] however it remains 'unclear how many South Africans access and receive rehabilitation services after sustaining a stroke, what rehabilitation is provided to them, how effective this rehabilitation is and what the implications are, of receiving inadequate, or even no, rehabilitation'.[26]

## Universal health coverage for people with stroke in SA

The proposed National Health Insurance (NHI) bill, currently being considered by the National Assembly of SA[31] focuses on primary healthcare as the foundation for universal health coverage (UHC), however the bill may not provide coverage for most age-related health conditions such as mobility impairments, and visual or auditory impairments.[32] UHC occurs where everyone receives essential services, according to their need and without financial hardship.[33] Essential services are defined as promotion, prevention, treatment, rehabilitation (including assistive technology) and palliative care. Key features of UHC include its human rights-focused underpinning and an integrated approach to health service delivery. It also importantly recognises the role that health system functioning plays in the realisation of UHC.[34] In recent years, SA has spearheaded radical healthcare policy reform to facilitate patient-centred and

accessible service delivery as outlined in the Healthcare 2030 plan of the National Department of Health.[26] This has mainly been actioned via a decentralised district level healthcare service delivery model and the development of an NHI scheme to curtail financial hardship experienced by users.[35] Moving towards UHC is a priority in SA and requires a strong and responsive healthcare delivery system to anchor effective and efficient service delivery. There is a crucial need for 'good leadership, stewardship, and management of health and related services to achieving health for all people'.[4] Louw *et al* further affirm that 'improving capacity across African healthcare settings is essential to ensure best practice health programmes to increase awareness of stroke and its causes, its early identification and acute management, and its rehabilitation'.[36]

In general, stroke care services in Africa are not adequately supported by governments, with limited support for either the development or implementation of national stroke policy frameworks, or limited provision and endorsement of evidence-based clinical practice guidelines.[37] Where policy frameworks or practice guidelines are available, the implementation processes may not be well supported by states,[38] which further limits best practice care for people with stroke.[37] In addition, uniquely African clinical practice guidelines for stroke care have not been well reported.[39 40] Practice guidelines that are appropriately adapted for local context are important, and consequently, the South African-contextualised Stroke Rehabilitation Guideline,[22] developed in collaboration with key stakeholders, including the South African Department of Health, were developed between 2017 and 2018. This guideline marked a strategic and collaborative increased focus on the importance and quality of stroke rehabilitation in the SA healthcare context. However, the update of such guidelines in at all levels of care remain a challenge.

Accessible, responsive and quality stroke care services within a strengthened local health system will contribute to UHC for people with stroke and their caregivers in SA. The extent to which UHC is achieved for people with stroke in SA is unknown and relatively little is known about the opportunities and challenges within the local health system to achieve UHC.

### Aim of this scoping review

The aim of this review is to explore the opportunities and challenges within the South African health system to facilitate the achievement of UHC for people with stroke. A scoping review method was chosen to address the broad nature of our question as well as identify knowledge gaps to inform further research. For the purpose of this scoping review, we will specifically explore the components and characteristics of the stroke-related health system as well as policies that potentially facilitate or hinder stroke care. We will draw on two frameworks: Health Systems Dynamics Framework (HSDF)[41] and WHO Framework on integrated people-centred health services (IPCHS)[42] to map the data from different

sources in a thematic content analysis. This information is required to adequately plan for the healthcare needs of people with stroke in the local SA context.

### METHODS

We will conduct this scoping review according to the approach recommended by Levac *et al*,[43] which includes five steps: (1) identifying the research question, (2) identifying relevant studies, (3) selecting the studies, (4) charting the data, and (5) collating, summarising and reporting the results. The review will be reported according to the Preferred Reporting Items for Systematic Reviews and Meta-Analyses extension for Scoping Reviews (PRISMA-ScR) guidelines using a modified PRISMA flow chart.[44]

### Analytical framework

Our review will be guided by an analytical framework adapted from the HSDF[40] and WHO Framework on IPCHS[41] to map and synthesise data from different sources in a thematic content analysis. The HSDF incorporates the WHO health system building blocks (1) service delivery; (2) health workforce; (3) information; (4) medical products, vaccines and technologies; (5) financing; and (6) leadership and governance, and highlights how values and principles drive the behaviour of people when making choices and engaging with processes of any health system. This HSDF framework offers an integrated view of a health system in that it acknowledges the social, economic, political context and determinants of health. The WHO IPCHS aims to support countries to achieve UHC by facilitating access to health services that are provided in a way that is coordinated around people's needs, respects their preferences, and are safe, effective, timely, affordable, and of acceptable quality. Our analytical framework will include all of the HSDF components and two components from the IPCHS: (1) re-orientation of care and (2) enabling environment, which are appropriate to the SA context and population (see figure 1).

### Step 1: defining the review objectives

In line with the purpose of scoping reviews, our approach is broad, with emphasis on studies that investigate any aspect of the healthcare system in regard to stroke care in SA. The definitions of the concepts that will be used for this review is provided in online supplemental file 1.

The review objectives are to:
1. Describe the health system-related factors such as governance, resources, community engagement and service delivery that support and guide stroke care in SA.
2. Describe the health system-related factors such as governance, resources, community engagement that limit the achievement of universal stroke care in SA.
3. Synthesise the findings to gain insights into the driving factors that will fundamentally bring required change to achieve universal stroke care in SA.

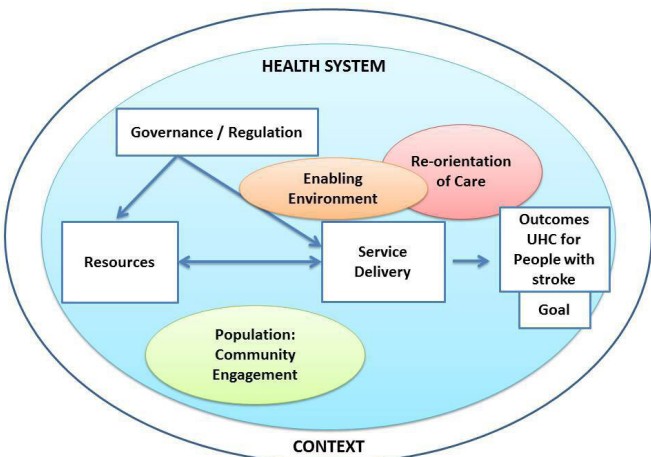

**Figure 1** Analytical framework (adapted from the Health Systems Dynamics Framework[41] and the WHO Framework on integrated people-centred health services components[42]). UHC, universal health coverage.

## Step 2: identifying relevant studies

### Search strategy

A comprehensive search, with the most appropriate search terms per database, will be conducted for each of the following databases: PubMed, EBSCOhost (including Medline and CINAHL), Scopus, Global Health and African-wide databases (including EBSCOhost Africa Wide, African Journals Sabinet and African Journals Online). The databases were chosen to ensure that all relevant literature was identified. An initial, limited search selection of relevant databases will take place, followed by an analysis of text words in the title and abstract, as well as the index terms used to describe the article. To ensure that all relevant studies are included, additional, peer-reviewed literature will be added by hand searching the reference lists of the articles that were initially included to ensure that articles have not been missed. The search strategy developed for the purpose of the proposed review is provided in online supplemental file 2.

Grey literature will be identified by using the WHO's OpenGrey and OpenDOAR Library, as well as sources such as Open Access Theses and Dissertations, Sabinet: Current and Completed Research, ProQuest Dissertations and Theses, Database of African Theses and Dissertations, and the Networked Digital Library of Theses and Dissertations to include research conducted on stroke care in SA. The websites of relevant government and service provider agencies will be searched to identify relevant South African stroke care policy documents and/or practice guidance documents. In addition, field experts will be contacted to identify additional relevant evidence regarding stroke care in SA. Saturation will be the point at which no new evidence is found to be included in the scoping review.

A Boolean search string has been developed through the systematic process of reviewing Medical Subject Heading terms on Medline. A key term search strategy will be employed using a variety of combinations of terms

for 'stroke', AND 'health system' OR 'Universal Health Coverage' OR 'clinical practice guideline' AND 'South Africa'. We will also conduct specific key term searches using each of the seven different components from the analytic framework (figure 1), combined with 'stroke' and 'South Africa'. We have conducted a pilot search in Google Scholar on the 9th of May 2020, to demonstrate the feasibility of answering our research question using a scoping review method. For the pilot search we have used the following combination of search terms 'stroke' AND 'South Africa' AND 'health system' OR 'Universal Health Coverage' OR 'clinical practice guideline' -cardiovascular -diabetes', which resulted in 2440 hits and a possible 100 publications were retrieved by title.

### Eligibility criteria

This scoping review will include primary and secondary research studies, policy documents, reports and clinical practice guidelines describing any aspect of the healthcare system for people with stroke in SA. The review will include qualitative, quantitative and mixed methods studies of any study design. Records published or made available between 2005 and 2020 will be accepted for inclusion. A national reform of the SA healthcare system was called on in 2009, thus this timeframe would ensure that we include all possible studies that focused on the functioning of the SA health system since the time of reform until the present moment.[45]

Records will be limited to those taking place in SA. Only records where the full text is available (in English or Afrikaans) will be included. This is especially because the research team possess these language skills and its contextually relevant. Finally, evidence will be interrogated as to whether they addressed at least one of the components of the analytical framework.

### Step 3: evidence selection

Following database search, retrieved articles will be screened in three stages. First, one reviewer will independently screen the titles and abstracts of identified studies to exclude publications that do not meet the inclusion criteria. A second reviewer will check the results for accuracy. The data will be managed with EndNote V.X8. The full-text article will be retrieved for review if the citation is deemed eligible by at least one reviewer. Two of these reviewers will independently assess each article against the inclusion criteria. Any discrepancies between the reviewers will be resolved by discussion, and a third reviewer will be consulted if necessary. A modified PRISMA that incorporates PRISMA-ScR reporting framework will be completed to summarise the study selection process.

### Step 4: data charting

Data extraction will be undertaken independently by three reviewers (S-MvN, GI-J, SK). Each reviewer will be allocated specific data item(s) to extract, a second reviewer will check the extracted data for accuracy. Before

**Table 1** Application of the key characteristics of the analytical framework

| Framework component | Description (data items) |
|---|---|
| Governance/ regulation | Healthcare policies at national or provincial levels; resource allocation policies; accountability; coordination and regulations. |
| Resources | Infrastructure (accessible; equipped and maintained); finance allocation and affordability; human resources (availability; distribution—occupation/specialisation, place of work, gender; graduates of health professionals); knowledge (clinical decision-making; clinical guidelines). |
| Service delivery | Service delivery models; health services and service providers (private/public, for-profit or not–for-profit, formal or informal, professional or non-professional, allopathic or traditional, remunerated or voluntary). |
| Context | Socioeconomic, technological, cultural, political and environmental environments. |
| Reorientation of care | Health promotion and patient education. New technologies (eHealth; shared electronic medical records; telemedicine; m-health). Coordination of health services; continuous healthcare, multiprofessional health teams; comprehensive health services and intersectoral coordination. |
| Enabling environment | Health workforce allocation/shortages; distribution of health workforce; workforce training; clearly defined roles and fair wages |
| Population: community engagement | Engaging and empowering individuals, families, communities and informal carers. Reaching underserved and marginalised communities. Common decision-making, self-efficacy of patients. |

commencing data extraction, the reviewers will first discuss the information to be extracted to ensure clarity of the data extraction process. Any discrepancies between the reviewers will be resolved by discussion, and a third reviewer will be consulted if necessary. A custom-designed form will be developed in Excel for data charting. Two independent reviewers will pilot the data charting form using a random sample of five included studies for consistency and required amendments will be agreed by consensus. We will modify the data extraction form as required based on feedback from the two reviewers, and the form amended at each stage where necessary. We plan to contact study authors in the case of unclear information and will make up to three attempts by email.

## Data items

The following general information will be extracted and tabulated from the included articles: author name(s), publishing journal, year of publication, region in SA, study population, the study aims/objectives/question, the setting, study-design and findings. In addition, the following data will be extracted according to the components of HSDF and the WHO's IPCHS framework components. To ensure consistency of the data extraction process, we will consider the seven components using the descriptions in table 1.

## Data synthesis and analysis

The analytical framework will guide the data synthesis and thematic analysis. We anticipate that the dataset will include different study designs, and therefore both descriptive statistics and narrative synthesis will be used. We will establish the strengths and opportunities that should be optimised in order to improve UHC of stroke services in SA as well as the weaknesses and threats which need to be minimised or eliminated.

## Availability of data and materials

All data generated or analysed during this study will be included in the published scoping review article.

## Patient and public involvement

No patients and public were involved for the purpose of this scoping review.

## Ethics and dissemination

Ethical approval is not required for this scoping review, as it will only include published and publicly available data. The findings of this review will be published in an open-access, peer-reviewed journal and we will develop an accessible summary of the results for website posting and stakeholder meetings.

**Author affiliations**
[1]Division of Physiotherapy, Department of Health and Rehabilitation Sciences, Stellenbosch University, Cape Town, South Africa
[2]SACDIR, Public Health Foundation of India, Gurgaon, India
[3]International Center for Evidence in Disability, London School of Hygiene and Tropical Medicine, London, UK
[4]Department of Global Health and Development, Faculty of Public Health and Policy, London School of Hygiene and Tropical Medicine, London, UK
[5]Disease Control Department, Faculty of Infectious and Tropical Diseases, London School of Tropical Health and Medicine, London, UK
[6]Division of Health Systems and Public Health, Department of Global Health, Faculty of Medicine and Health Sciences, Stellenbosch University, Cape Town, South Africa

**Correction notice** The article has been corrected since it is published. The affiliations 5 and 6 have been updated.

**Contributors** S-MvN and GI-J drafted and revised the protocol with suggestions from TS, JW, SK, SF, RE and QAL who reviewed the protocol and provided feedback on the draft. S-MvN, SF and GI-J in consultation with the other authors constructed the search. All authors read and approved the final protocol.

**Funding** This research was commissioned by the National Institute for Health Research (NIHR) Global Health Policy and Systems Research Development Award using UK aid from the UK government. Grant number NIHR130180.

**Disclaimer** The views expressed in this publication are those of the author(s) and not necessarily those of the NIHR or the Department of Health and Social Care.

**Competing interests** None declared.

**Patient and public involvement** Patients and/or the public were not involved in the design, or conduct, or reporting, or dissemination plans of this research.

**Patient consent for publication** Not required.

**Provenance and peer review** Not commissioned; externally peer reviewed.

**ORCID iDs**

Sureshkumar Kamalakannan http://orcid.org/0000-0003-4407-7838
Tracey Smythe http://orcid.org/0000-0003-3408-7362

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
