## [Reviewer comments · BMJ Open]

ARTICLE DETAILS

TITLE (PROVISIONAL)	Achieving Universal Health Coverage for people with stroke in South Africa: Protocol for a scoping review
AUTHORS	Sjan-Mari, van Niekerk; Gakeemah, Inglis-Jassiem; Sureshkumar, K; Fernandes, Silke; Webster, Jayne; English, Rene; Smythe, Tracey; Louw, QA

VERSION 1 – REVIEW

REVIEWER	Daudet Tshiswaka University of West Florida
REVIEW RETURNED	11-Jul-2020

GENERAL COMMENTS	Abstract Line 8, please write the UHC acronym in parentheses next to its explanation. No need to put it in parentheses on line 42. Introduction Please put DALYs in parentheses and spell it out the first time you use it. While I appreciate mentioning COVID-19 on lines 42-45, I have the impression that this sentence is a bit out of context. Do the underlying conditions refer to stroke survivors? Eligibility criteria How will you deal with articles that have been written in Afrikaans? Figure 1 What does PWS stand for?
--

REVIEWER	Prof Jeyaraj D Pandian Dean and Professor of Neurology Christian Medical College Ludhiana Punjab, India 141008
REVIEW RETURNED	03-Aug-2020

GENERAL COMMENTS	This is an important review addressing the gaps in Universal health coverage for stroke in South Africa. This is a methodology paper. I would suggest to reduce the introduction section. Modified PRISMA chart could be included.
--

VERSION 1 – AUTHOR RESPONSE

Reviewer 1			
Abstract			
5	Line 8, please write the UHC acronym in parentheses next to its explanation. No need to put it in parentheses on line 42.	Revised as suggested	The line numbers were not matching but we have spotted what the reviewer was suggesting and have made the changes . Highlighted in Yellow
Introduction			
6	Please put DALYs in parentheses and spell it out the first time you use it.	Revised as suggested	Highlighted in Yellow
7	While I appreciate mentioning COVID-19 on lines 42-45, I have the impression that this sentence is a bit out of context.	Revised the sentence to fall in line with the context as suggested	Highlighted in Yellow
8	Do the underlying conditions refer to stroke survivors?	Yes. We have revised the sentence as suggested	Highlighted in Yellow
9	Please include the definitions in a table or in a supplementary file.		
Eligibility criteria			
10	How will you deal with articles that have been written in Afrikaans?	Three authors of this review are fluent in Afrikaans. We have added a sentence about this in the manuscript	Highlighted in Yellow
11	Figure -1		
12	What does PWS stand for?	Persons with stroke – Revised for clarity	Highlighted in Yellow

Reviewer - 2			
Introduction			
13	I would suggest to reduce the introduction section	We have removed the definition Section and moved it as a Supplementary section. Cutting down about 500 words. Authors decided to report the background in details but strictly following the word count recommendations. Hence we would like keep the existing Introduction section as it is.	Supplementary file on definitions added
Methods / Evidence Synthesis			
14	Modified PRISMA chart could be included.	Revised as suggested	Highlighted in yellow at the evidence synthesis and methods section